# A Protein-Based Material from a New Approach Using Whole Defatted Larvae, and Its Interaction with Moisture

**DOI:** 10.3390/polym11020287

**Published:** 2019-02-08

**Authors:** Nazanin Alipour, Björn Vinnerås, Fabrice Gouanvé, Eliane Espuche, Mikael S. Hedenqvist

**Affiliations:** 1KTH Royal Institute of Technology, School of Engineering Sciences of Chemistry, Biotechnology and Health, Fibre and Polymer Technology, SE-100 44 Stockholm, Sweden; nalipour@kth.se; 2SLU Swedish University of Agricultural Sciences Department for Energy and Technology, PO Box 7032, SE-750 07 UPPSALA, Sweden; bjorn.vinneras@slu.se; 3UMR CNRS 5223, Ingénierie des Matériaux Polymères, 15, Bd. André Latarjet, Univ Lyon, Université Lyon 1, 69622 Villeurbanne Cedex, France; fabrice.gouanve@univ-lyon1.fr (F.G.); eliane.espuche@univ-lyon1.fr (E.E.)

**Keywords:** protein, larva, plastic, lipid, Black Soldier fly, moisture sorption

## Abstract

A protein-based material created from a new approach using whole defatted larvae of the Black Soldier fly is presented. The larvae turn organic waste into their own biomass with high content of protein and lipids, which can be used as animal feed or for material production. After removing the larva lipid and adding a plasticizer, the ground material was compression molded into plates/films. The lipid, rich in saturated fatty acids, can be used in applications such as lubricants. The amino acids present in the greatest amounts were the essential amino acids aspartic acid/asparagine and glutamic acid/glutamine. Infrared spectroscopy revealed that the protein material had a high amount of strongly hydrogen-bonded β-sheets, indicative of a highly aggregated protein. To assess the moisture–protein material interactions, the moisture uptake was investigated. The moisture uptake followed a BET type III moisture sorption isotherm, which could be fitted to the Guggenheim, Anderson and de Boer (GAB) equation. GAB, in combination with cluster size analysis, revealed that the water clustered in the material already at a low moisture content and the cluster increased in size with increasing relative humidity. The clustering also led to a peak in moisture diffusivity at an intermediate moisture uptake.

## 1. Introduction

NASA reported that 2016 was globally the warmest year for surface temperature since modern record-keeping started in 1880 [1]. It is clear that the amount of greenhouse gases generated has to decrease. Even though the burning of fuel is the major source of CO_2_ generation, the burning of oil-based plastics also contributes. Replacing commodity thermoplastics with corn-based bioplastics, which is one of the most realistic options in the near future, can reduce the emission of greenhouse gases by 25% [2]. Besides plants/crops, animal-based renewable resources are also available for making plastics as a more environment-friendly alternative to fossil-based oil [3]. These are proteins such as collagen/gelatine, casein/whey, keratine, fibroin and fish myofibrillar proteins [4] and polysaccharides such as chitin/chitosan [5]. A hitherto unexplored animal source for plastic production is larvae biomass. Larvae of the Black Soldier fly (BSF) (*Hermetia illucens*) are an interesting option for turning food waste and other organic matter into protein, lipid and chitin [6,7,8]. The BSF larvae are efficient eaters and grow rapidly. They can convert from 15% up to 35% of initial volatile solids (organic content) into their own biomass, depending on the substrate composition [9,10,11,12]. Mature larva biomass consists mainly of protein and lipid, with the remainder being (to a large extent) exoskeletal chitin/calcite material [13,14]. Because of their high concentration of lipid and proteins with essential amino acids, BSF larvae have been considered as feed for animals [15,16,17,18]. Although this is the primary application for BSF larvae, larvae may not be accepted as animal feed due to contaminants such as chemical pollutants, e.g., perfluoroctanesulfate (PFOS), or heavy metals [19,20,21,22,23], and in this context it is of interest to find alternative uses. Since the BSF larvae contain a large amount of protein, it is interesting to see whether it is possible to produce materials from the whole larvae. Whereas the production of many bio-based materials (e.g., poly(lactic acid), thermoplastic starch, sugarcane polyethylene and soy protein) may compete with the production of human food [24], protein materials from larvae grown on food waste do not.

In order to obtain a more cohesive material, the lipid has to be removed. This creates a side-stream product, with a potential for wax, oil and biodiesel production [25,26]. It is also possible to extract the chitin from the larvae for the production of chitosan materials. It has been shown that this chitin is similar in quality and properties to that from insects [8].

In the present study, we have investigated the secondary structure and properties of a protein material produced from ground defatted plasticized BSF larvae, especially its interactions with water/moisture. This is the first study using whole defatted larvae for plastic material production. The biggest hurdle for replacing oil-based plastics with protein materials is their moisture sensitivity. It dictates to a large extent the protein material properties [27]. It is thus important to investigate these interactions in detail for new candidates. The protein secondary structure is also important to reveal since it does also dictate to a large extent the material properties.

## 2. Experimental

### 2.1. Materials

The larvae used for material production were obtained in a fly larva composting unit [11]. The larvae were fed with a combination of pig manure, dog food (Puppy Original, Pro Plan, Purina) and human feces (4:4:2 in weight, 28.7 ± 1.2 total solids) [11]. The larvae at the sixth instar (pre-pupae) were separated from the compost and placed in plastic bags in a freezer at −25 °C during at least one week for inactivation, dried at 105 °C for 14 h and subsequently cooled to room temperature and stored at ambient water conditions (larva moisture content: ~10 wt.%). The whole larvae (ca. 100 g) were cryo-ground using a Retsch 5657 HAAN grinder (Germany). The lipid was extracted by adding five parts n-hexane to one part larva powder (20 g/4 g). The mixture was stirred with a magnetic stirrer at 60 °C for 3 h and then centrifuged for 5 min at 4000 rpm, after which the supernatant (n-hexane and dissolved lipid) was decanted. The defatted powder was then left in an oven at 60 °C overnight to ensure that the remaining n-hexane evaporated. After drying, the powder was mixed manually with the glycerol plasticizer (70/30 *w*/*w*). Glycerol has been shown to be one of the most effective plasticizers for proteins [28] and the glycerol content used here has been used favorably on other protein-based materials, including soy protein [29], casein [30], whey protein [31], rapeseed protein [32], silk/fibroin [33], gelatin [34], zein [35], wheat gluten [36,37] and protein from oil crops such as *Crambe abyssinica* and *Brassica carinata* [38]. The mixture was compression-molded into plates/films using steel frames at 100 °C under a compressive load of 320 kN for 5 min (Figure 1). The heating was then stopped, and the sample was cooled to room temperature without releasing the pressure. The thinnest samples were used in the present study in order to obtain reasonably short measurement times in the Dynamic Vapor Sorption equipment. Different procedures for making the material were tested, including an n-hexane/powder ratio of 3/1 and two-step centrifugation, but the procedure given above was found to be the best with respect to removing most of the lipid efficiently while limiting the number of processing steps.

### 2.2. Density and Extraction in Liquids

The density of the material was measured under ambient conditions from the volume and weight of pieces of cut film. The extraction was performed by immersing the film in milliQ water, ethanol or n-heptane at room temperature for three days.

### 2.3. Infrared (IR) Spectroscopy

The sample, dried for at least two days in a desiccator with silica gel, was investigated using a Perkin-Elmer Spotlight 400 Fourier Transform Infrared (FTIR) imaging system equipped with a germanium attenuated total reflection (ATR) crystal. The IR absorption spectrum was recorded in the ATR image mode in the regions 750 cm^−1^ and 4000 cm^−1^ based on 16 scans and a resolution of 4 cm^−1^. The IR spectrum was resolved into nine Gaussian peaks in the amide I band (1587 cm^−1^ to 1700 cm^−1^) using fixed peak positions, and assignments of these according to Cho et al. [39]. Before the peak fitting, the raw IR data was first deconvoluted using an enhancement factor (γ) of 2 and a smoothing filter of 70%. The first fit was the total amide I absorbance, which included the absorbance from the amide in the chitin. The total amide I absorbance spectrum was then reduced in size with respect to the measured chitin absorbance spectrum (not shown), assuming that the absorbance was a linear function of the mass content of the different components (protein, chitin) in the material. A new peak fitting was performed on the reduced spectrum starting with the fitted parameters from the total amide I spectrum.

### 2.4. Thermogravimetry and Calorimetry

The mass and heat flow were measured on a Mettler-Toledo TGA/DSC 1 Stare system using either air or nitrogen as purge gas (flow rate: 50 mL/min). The samples (~20 mg) were heated at a rate of 10 °C/min from 25 °C to 800 °C.

### 2.5. Dynamic Vapor Sorption (DVS)

A dynamic vapor sorption analyzer, DVS Advantage by Surface Measurement Systems, was used to obtain the water sorption isotherms of the thin film. The partial vapor pressure was controlled by mixing dry and water-saturated nitrogen using electronic mass flow controllers at 25 °C. The initial weight of the sample was approximately 15 mg. The sample was pre-dried in the DVS Advantage by exposing it to dry nitrogen until the equilibrated dry mass (*m_0_*) of the sample was obtained. The partial vapor pressure (*p*) was then established within the apparatus and the mass of the sample (*m_t_*) was recorded as a function of time. The mass of the sample at equilibrium (*m_eq_*) was considered to have been reached when change in mass with time (*dm*/*dt*) was lower than 2 × 10^−4^ mg/min for at least five consecutive minutes. The vapor pressure was then increased in steps of 0.1 in water activity (*a_w_*) up to an activity of 0.9. The value of the mass gain at equilibrium (*M*), defined as $\frac{m_{eq}-m_{0}}{m_{0}}$, for each water activity, was then plotted to give the water sorption isotherm. The precision of the values of the mass gain at equilibrium was estimated to be better than 3%. The water diffusivity was calculated from the half-time (*t*_0.5_) to reach the saturation water content for each step change in water activity [40]:(1)$$D=\frac{0.049}{\frac{t_{0.5}}{l^{2}}},$$
where *l* is the sample thickness.

## 3. Results and Discussion

### 3.1. Larva Constituents

The raw protein content of the larvae was 43.5 ± 1.6 wt.% of the dry mass according to Kjeldahl (N × 6.25, NMKL 6:2003). The sum of the contents of each amino acid yielded a protein content of 39.2 ± 2 wt.%. The difference, 4.3%, was considered to be due to chitin [41]. The amino acids with the highest content were the essential amino acids aspartic acid/asparagine and glutamic acid/glutamine, each amounting to ca. 10 wt.% of the total amino acid content (Table 1). Other amino acids present in larger amounts were the non-essential amino acid tyrosine (9 wt.%) and the essential amino acids leucin, lysine and valine (~7 wt.% each). The content of the sulfur-containing and disulfide forming cystine/cysteine was less than 1 wt.%.

The lipid extracted with n-hexane was both liquid and solid and amounted to ca. 30 wt.% of the whole larvae (Figure 2). It consisted of 72 wt.%, 16 wt.% and 10 wt.% of saturated, mono-unsaturated and poly-unsaturated fatty acids, respectively (Table 2). These values are similar to those reported earlier on pre-pupae oil despite feeding the larvae with a different source (70 wt.%, 15 wt.% and 12.5 wt.%). As previously reported for the pre-pupae oil, the main fatty acids were lauric (48 wt.% of the total fatty acid content), palmitic (14 wt.%), oleic (13 wt.%), linoleic (8 wt.%) and myristic acid (7 wt.%) (Table 2). The remaining dry matter of the larvae consists of minerals/ash, fibers, organic compounds, carbohydrates and chitin [18,42].

### 3.2. Protein Structure

The conformational structure of the protein was determined with IR spectroscopy. Figure 3 shows the protein absorbance in the Amide I region after subtraction of the chitin absorbance in the same region. The positions and origins of the nine resolved peaks after deconvolution and fitting are given in Table 3. The content of strongly hydrogen-bonded β-sheets was 44% (peaks 1 and 2), which is indicative of a large amount of aggregated protein. It is similar to a highly aggregated wheat gluten sample (42.2%) with the same glycerol content that has undergone an injection molding cycle at high temperature (180 °C) (Table 3). The content of α-helices and unordered proteins (peaks 4 to 6) was 33%, higher than that of the wheat gluten material (21%). The content of weakly hydrogen-bonded β-sheets (peaks 3 and 8, not specifically associated with highly aggregated protein) was 7%, lower than that of the wheat gluten material (12%). Finally, the content of β-turns (peaks 7 and 9) was lowest for the larva protein (16%, wheat gluten: 25%). A highly aggregated protein usually means a protein with high cohesion. However, this is also a matter of the purity of the protein material. The present material did not have the cohesion expected from the high aggregation, which is due to, as in the case reported for rapeseed flour/cake materials [43], the lower protein purity. Despite the defatting, a further purification step is needed for improved cohesion or, alternatively, a crosslinking of the protein component. However, every purification step will increase the price of the material so there is a trade-off regarding protein purity.

### 3.3. Thermogravimetry and Calorimetry

In Figure 4, the mass change and heat flow during heating in both air and nitrogen are shown. The loss in mass up to ca. 100 °C is due to evaporating water and further on the evaporation of glycerol (endothermal processes). Beyond ca. 200 °C, the net heat flow is exothermal and due to material degradation (oxidation in air and pyrolysis in nitrogen). The residual char at 800 °C is, as expected, higher in the pyrolysis case than in the case of oxidation. However, it is interesting to note that in both cases, the residual char is substantial. This is beneficial when it comes to the flame retarding properties of plastics.

### 3.4. Dynamic Water Vapor Sorption

Figure 5 shows the water sorption isotherm for the larva material and, for comparison, those of a wheat gluten sample with a similar glycerol content (27%). The latter was prepared with the same technique as for the larva material (compression molding) and at a similar temperature (80 °C and 120 °C). The shapes of the sorption isotherms (BET type III [43]) are similar, but the water uptake is slightly lower in the larva material at a high water activity (*a_w_*), which is beneficial for plastics use. The shape of the sorption isotherm (Figure 5) and the mathematical description of the underlying sorption mechanisms yield information about the sorption mode and the interactions involved in the sorption process. There is a noticeable absence of a Langmuir-type sorption and instead a Henry’s law type of uptake up to a water activity of ca. 0.2 (20% relative humidity (RH)), the expected behavior for a glycerol-plasticized bio-based polymer with a moderate-to-high plasticizer content. For these materials, fewer free sorption-specific sites are expected due to the formation of glycerol–polymer hydrogen bonds. Several models are available in the literature for describing water vapor sorption in polymers. The Guggenheim, Anderson and de Boer (GAB) equation is one of the more frequently used equations for the fitting of BET type II and BET type III isotherms [44]. These isotherms are observed for both natural and synthetic polymers [44,45]. The GAB equation is based on the assumption of localized physical sorption in multilayers with no lateral interactions. The first molecules are in strong interaction with the polymer chains and form a monolayer. The subsequent layers of water molecules interact to a lower degree with the sorbent and the range in energy levels is between that of the monolayer and that of the bulk liquid. The GAB model, predicting the mass uptake of water (*M*), is expressed as [45]:(2)$$M=M_{m}\frac{C_{G}\;K\;a_{w}}{\left(1-K\;a_{w}\right)\left(1+\left(C_{G}-1\right)K\;a_{w}\right)},$$
where *M*_m_ is the amount of water in the monolayer, and a measure of the availability of active sites on the polymer for water molecules. *C*_G_ is the Guggenheim constant, which is a measure of the strength of binding of water to the primary binding sites. *K* is a correction factor for the difference in properties of the multilayer molecules and of those in the bulk liquid [45]. To evaluate the goodness of fit of this model (obtained by the Tablecurve 2D software (Jandel Scientific, San Rafael (CA)), the mean relative percentage deviation modulus (*MRD*) was used, defined as:(3)$$MRD\;\left(\%\right)=\frac{100}{N}\sum_{i=1}^{N}\frac{\left|m_{i}-m_{\mathrm{pi}}\right|}{m_{i}},$$
where *m*_i_ is the experimental value, *m*_pi_ is the predicted value, and *N* is the number of experimental measurements. The parameters obtained from the GAB model and the *MRD* for the larva material and for compression-molded wheat gluten samples are reported in Table 4. An *MRD* value below 10% indicates a good fit for any practical purpose [45]. The low *MRD* value for the fit of Equation (2) to the experimental data in Figure 5 (1–7%), indicates that the model was suitable and gave an accurate description of the experimental sorption isotherm for all three samples. The *M*_m_ values were similar to those reported for other materials/biopolymers [46], indicating that the amounts of water involved in the monolayer formation were similar. The *C*_G_ and *K* values were also similar for all three samples, suggesting that the strength of the bonds between the water and the primary binding sites on the protein chain (*C*_G_) and the difference in the properties of the bulk water and the water involved in the multilayers were similar. It has been shown for starch that the addition of a plasticizer leads to a decrease in *M*_m_ and *C*_G_ and an increase in *K* [45], which may be interpreted as indicating that in the presence of a plasticizer, there are fewer available sorption sites for water molecules on the polymer chain (*M*_m_ decreases) and that those available provide a lower binding strength with the water (lower *G*_G_). Consequently, the apparent sorption energy decreases in the multilayers (increasing *K*).

The large increase in uptake observed for BET type II and BET type III isotherms at high activities is generally explained as being due to a clustering phenomenon. Zimm and Lundberg [48] developed a method based on statistical mechanics that provides information on this phenomenon from the shape of the sorption isotherm. Their method gives an interpretation of the solution thermodynamic behavior in geometric isotherms, using the equation:(4)$$\frac{G_{11}}{V_{1}}=-\left(1-\phi_{w}\right)\left(\frac{\partial (\frac{a_{w}}{\phi_{w}})}{\partial a_{w}}\right),$$
where *G*_11_ is the cluster integral, *V*_1_ is the partial molecular volume of the water molecules and *φ*_w_ is the volume fraction of the water molecules. A value of *G*_11_/*V*_1_ equal to -1 indicates that water molecules dissolve randomly in the polymer matrix, whereas higher values (*G*_11_/*V*_1_ > −1) indicate that the concentration of water in the neighborhood of a given water molecule is greater than the average concentration of water molecules in the polymer. It has been successful in interpreting the different modes of water sorption in proteins, from strong-binding to diffuse swelling. The quantity *G*_11_*φ*_w_/*V*_1_ is the mean number of molecules in excess of the mean concentration of water in the neighborhood of a given water molecule. Thus, the mean cluster size (*MCS*) is given by:(5)$$MCS=1+\left(\frac{\phi_{w}G_{11}}{V_{1}}\right).$$

In the case of water transport, the Zimm and Lundberg theory has often been coupled with the approaches that provide a mathematical description of the water sorption isotherms. As a result, *MCS* values can be expressed from the GAB parameters, considering the equation [49]: (6)$$MCS_{\mathrm{GAB}}=\frac{\rho^{2}}{M^{2}\left(1+\frac{\rho }{M^{2}}\right)}\left[1-\frac{M}{M_{m\;}C_{G}}\left(1-2\;K\;a_{w}\left(C_{G}-1\right)-2+C_{G}\right)\right],$$
where *ρ* = *ρ*_w_/*ρ*_p_, *ρ*_w_ and *ρ*_p_ are respectively the density of water (998 kg/m^3^, engineeringtoolbox.com) and the density of the polymer (1400 kg/m^3^, determined here from known geometry and weight), *M*_m_, *C*_G_, *K* are the three GAB parameters and *M* is the mass gain. The larva and wheat gluten materials showed a similar behavior (Figure 6). Water clustering occurred already at a low activity (small deviation from the *MCS* = 0 line around 0.2) and the *MCS* value was equal to 2.2 and 2.3/2.4 respectively for the larva and wheat gluten materials at a water activity of 0.9. The data for starch shows that, in the presence of glycerol, water clustering/auto-association is favored and begins at a lower water activity (0.48 (48% RH)) than when the plasticizer is absent (0.79 (79% RH)) [45]. This can be explained, as mentioned above, by less available active sites on the polymer molecule in the presence of the plasticizer.

The water diffusivity values are shown in Figure 7. As expected, the plasticizing effect of water led to an increase in the water diffusivity with increasing water activity. However, beyond a water activity of 0.35–0.4, the diffusivity decreased due to increasing water clustering. In Figure 7, the diffusivity data are also given for the wheat gluten material with a similar glycerol content. A peak in diffusivity was also observed, but it occurred at a significantly higher water activity than for the larva material, despite the similarity in the *MCS*–water activity data. Roy et al. [50] reported that the water diffusivity increased monotonically over the measured relative humidity range (6–80%) for a cast wheat gluten film with a glycerol content similar to that of the larva material. The water diffusivity–activity relationship is evidently sensitive to how the materials are produced. It is clear, however, that in the two wheat gluten materials referred to here, the water diffusivity was higher than that of the larva material, regardless of the actual water activity. Protein materials suffer from changes in mechanical properties during variations in the relative humidity. It is therefore beneficial to have a lower water diffusivity since the response to a change in relative humidity will be slower and property changes will smoothen out over time to a greater extent.

Below a water activity of 0.3, the water diffusivity could be fitted to an exponential expression often applied to describe the solute concentration-dependent diffusivity in swelling systems [51,52] (Figure 8):
(7)$$D\left(M\right)=D_{\mathrm{co}}e^{\alpha M},$$
where *D*_co_ (= 5.4 × 10^−9^ cm^2^/s) is the zero-concentration diffusivity and *α* (= 18.5 g material/g water) is the plasticization power. The equation describes the effects on the solute/water diffusivity of increasing free volume/molecular mobility due to the uptake of the solute molecules. The plasticization power describes the size of this effect. On the other hand, the region above a water activity of 0.3 could be fitted to [51]:(8)$$D\left(M\right)=D_{\mathrm{agg}}e^{-\beta M},$$
where *D*_agg_ (= 1.6 × 10^−8^ cm^2^/s) is the limit diffusion aggregation/clustering coefficient and *β* (= 4.6 g material/g water) is the anti-plasticization coefficient (Figure 8). This equation describes the decrease in water diffusivity with water content. As a consequence of that, the water molecules start to cluster and diffuse as larger aggregates. A rough estimate of the concentration dependence of the water diffusivity data for the wheat gluten material indicates that both *D*_co_ and *D*_agg_ are significantly higher for the larva material, but both *α* and *β* are lower (lower water plasticization and lower anti-plasticization efficiency in the wheat gluten cases), despite similar water cluster size/water activity trends (Figure 6). The break point between the two mechanisms occurs at a higher water activity/concentration in the case of the wheat gluten (Figure 7).

To reveal the interactions with polar and non-polar liquid molecules, the film was immersed in milliQ-water, ethanol and n-heptane. After immersion in water and ethanol for at least 43 h and drying under ambient conditions for 26 h, the samples had lost almost half the mass (48% (water) and 44% (ethanol) of the initial mass). The total (δ), polar (δ_p_) and hydrogen-bonding (δ_H_) Hansen solubility parameters are greater for water (δ = 47 MPa^0.5^, δ_p_ = 16 MPa^0.5^ and δ_H_ = 42.3 MPa^0.5^) than for ethanol (δ = 26.5 MPa^0.5^, δ_p_ = 8.8 MPa^0.5^ and δ_H_ = 19.4 MPa^0.5^) [53], which indicates the strong hydrophilicity of the larva protein film. With n-heptane (δ = 15.3 MPa^0.5^) [53], which has zero polar and hydrogen-bonding solubility parameters, no extraction occurred. The third dispersion (non-polar) Hansen solubility parameter was similar for all three liquids (δ_d_ = 15.6 MPa^0.5^ (water), 15.8 MPa^0.5^ (ethanol) and 15.3 MPa^0.5^ (n-heptane)) [53].

## 4. Conclusions

It has been shown here that it is possible to make a protein-based material by compression molding defatted larvae of the Black Soldier fly. This is a viable option, especially if the converted biomass cannot be used for animal feed due to polluted larvae. The conversion of food and organic waste to protein material through larvae is probably also a more economically/commercially interesting option than producing animal feed, since these materials are in general higher value products. There is also an interesting byproduct in the large amount of lipid extracted during the production of the protein material, which may find a use in bio-lubricants and biofuel. IR spectroscopy showed the presence of a highly aggregated structure and the behavior of the material in moist environments has also been investigated. Moisture is almost always a critical factor for the applications of protein materials, and it is therefore important to clarify the interactions between the material and moisture. The protein material with a glycerol plasticizer is highly hydrophilic, as shown by the large amount of the material extracted in water and the high moisture uptake shown by the sorption isotherm. In future work, the mechanical properties of the protein material under moist conditions is intended be investigated. Based on the results of this work, it is clear that the type of protein material investigated here can only find use in drier climates/indoor applications, separated from liquid water, or in applications where the material is in contact with fatty substances. Possible future applications could be as covers of lamps, covers of electrical parts/switches and disposable plastic products.

## Figures and Tables

**Figure 1 polymers-11-00287-f001:**
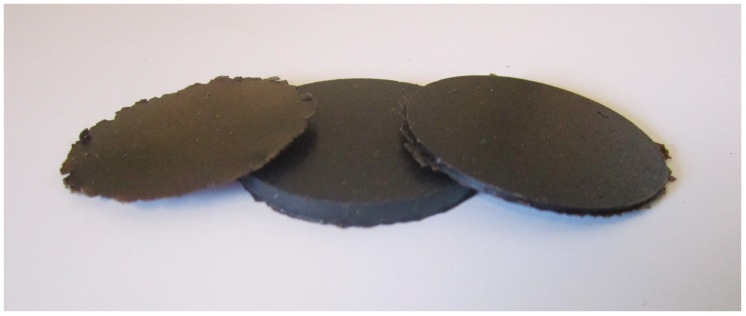
Compression-molded plates/films of the larva material (diameter: 35 mm, thickness from left to right: 180 µm, 3 mm and 1 mm).

**Figure 2 polymers-11-00287-f002:**
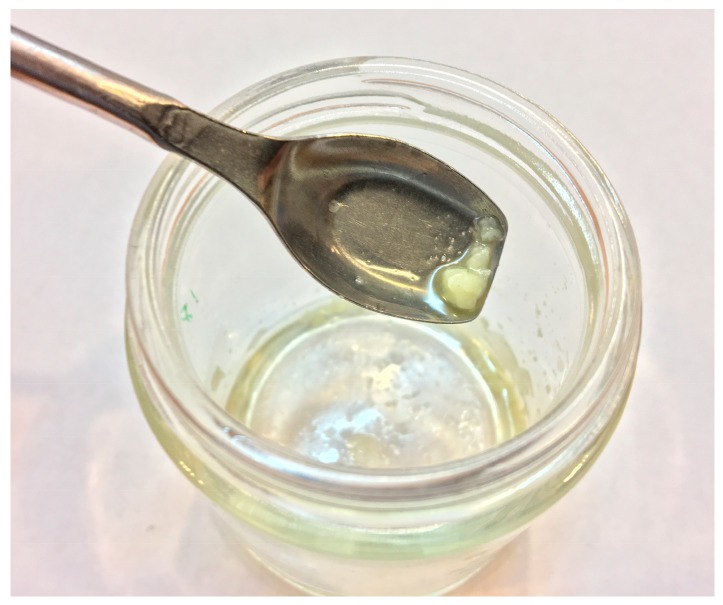
Lipid extracted with n-hexane.

**Figure 3 polymers-11-00287-f003:**
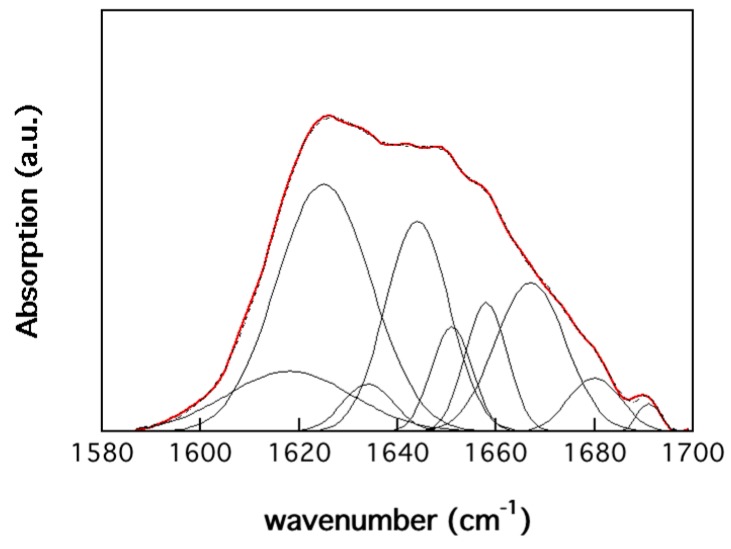
IR absorbance in the amide I region. The red and dashed curves correspond to the experimental and fitted curves, the latter made up of nine decomposed curves (Table 3).

**Figure 4 polymers-11-00287-f004:**
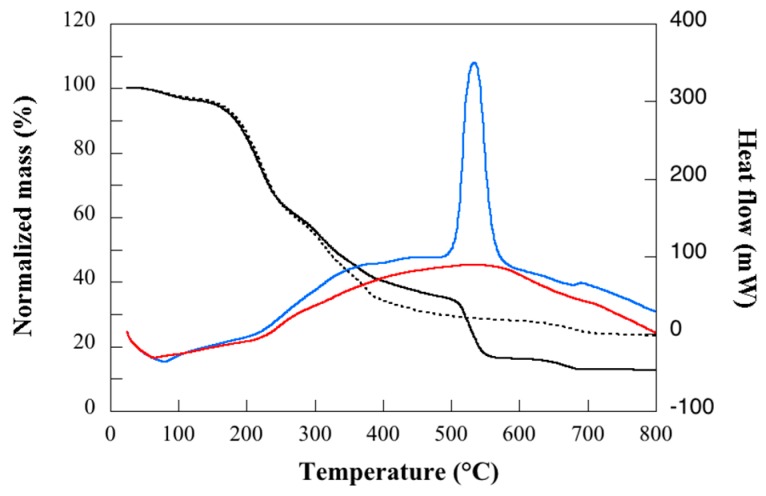
Mass and heat flow as a function of temperature. Black solid and broken curves refer to mass in air and nitrogen, respectively, and blue and red curves refer to heat flow in air and nitrogen. Exothermal heat flow is shown in positive numbers.

**Figure 5 polymers-11-00287-f005:**
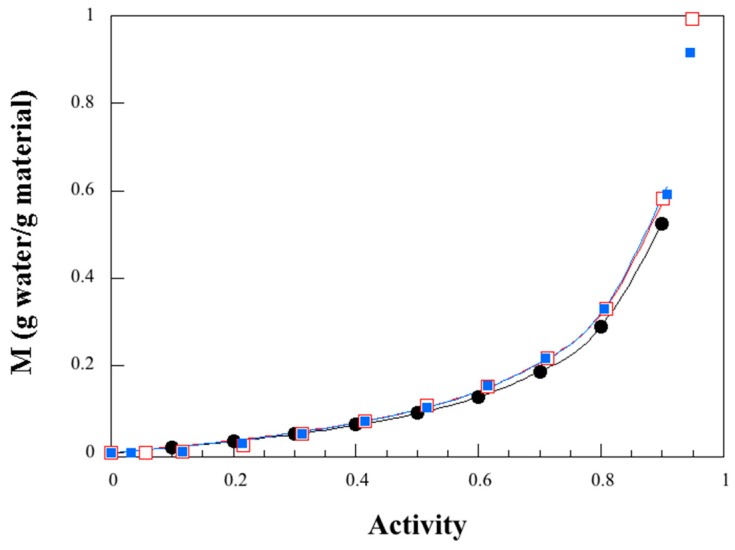
Water uptake as a function of water activity. Experimental data are indicated by filled circles (larva protein), unfilled squares (wheat gluten 80 °C sample) [47] and filled squares (wheat gluten 120 °C sample) [47], and the curves correspond to best fits of Equation (2).

**Figure 6 polymers-11-00287-f006:**
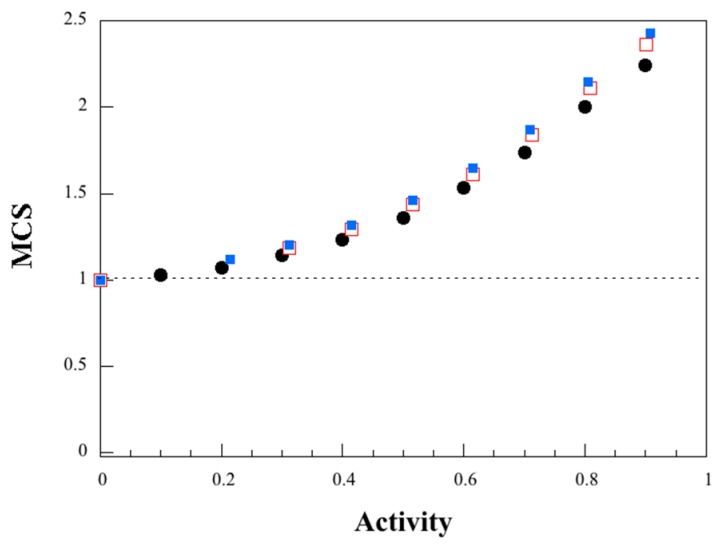
Mean cluster size (MCS) as a function of water activity; filled circles (larva protein), unfilled squares (wheat gluten 80 °C sample) and filled squares (wheat gluten 120 °C sample), calculated using Equation (6).

**Figure 7 polymers-11-00287-f007:**
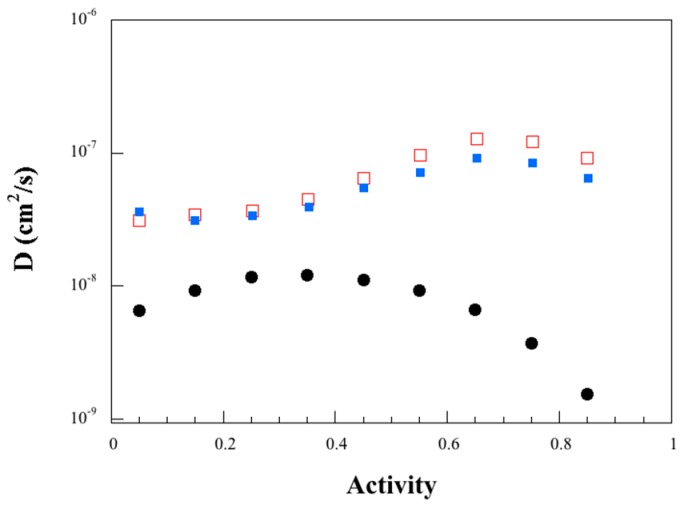
Water diffusivity as a function of water activity; filled circles (larva protein), unfilled squares (wheat gluten 80 °C, average values) [47] and filled squares (wheat gluten 120 °C, averages) [47].

**Figure 8 polymers-11-00287-f008:**
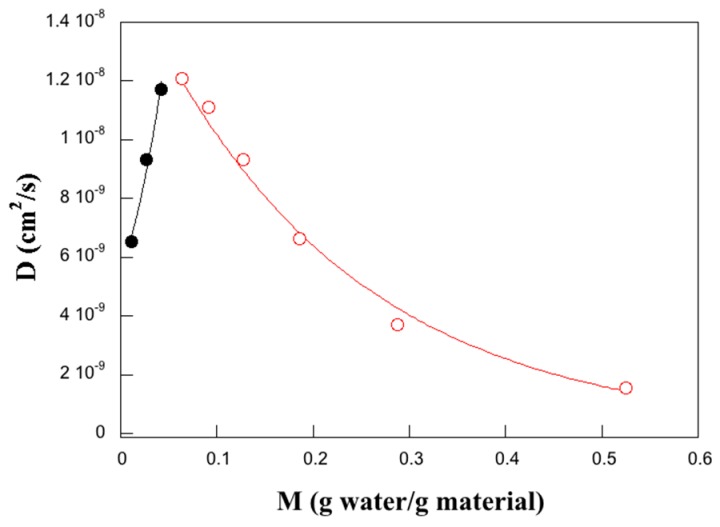
Water diffusivity as a function of water concentration; the black curve corresponds to the best fit using Equation (7) (R = 0.989) and the red curve corresponds to the best fit using Equation (8) (R = 0.997).

**Table 1 polymers-11-00287-t001:** Relative amino acid content ^a,b^.

Essential Amino Acids (wt.%)		Non-Essential Amino Acids (wt.%)	
Arginine	5.6 ± 0.2	Alanine	6.3 ± 0.1
Histidine	3.7 ± 0.3	Asparagine/Aspartic acid	10.3 ± 0.3
Isoleucine	5.1 ± 0.2	Cystine/cysteine	0.6 ± 0
Leucine	7.6 ± 0.2	Glutamine/Glutamic acid	11.1 ± 0.1
Lysine	7.1 ± 0.1	Glycine	6.3 ± 0.1
Methionine	2.0 ± 0.1	Proline	5.6 ± 0.1
Phenyl alanine	4.2 ± 0.5	Serine	4.5 ± 0.2
Threonine	4.2 ± 0.2	Tyrosine	8.9 ± 1.1
Valine	6.9 ± 0.2		

^a^ Relative to the total amino acid content. The ± values are standard deviations based on four samples. For detailed reports, see Appendix A. ^b^ The contents of hydroxyproline (non-essential) and ornitin (essential) were not included in the table because of very low values.

**Table 2 polymers-11-00287-t002:** Fatty acid composition (wt.% of total fatty acid content).

	Present ^a^	Reference ^b^
Capric acid (C10:0)	1	-
Lauric (C12:0)	48	45
Myristic (C14:0)	7	8
Palmitic (C16:0)	14	14
Stearic (C18:0)	2	2
Palmitoeic (C16:1 n-7)	2	2
Oleic (C18:1 n-9)	13	12
Linoleic (C18:2 n-6)	8	10
α-linolenic (C18:2 n-3)	2	0.1
Saturated fatty acids	72	70
Monounsaturated fatty acids	16	15
Polyunsaturated fatty acids	10	12.5

^a^ Values in the present study. For detailed report, see Appendix A. ^b^ In BSF pre-pupae oil from Black Soldier Flies (BSF) fed with food waste [14].

**Table 3 polymers-11-00287-t003:** Protein structure.

Peak	Position (cm^−1^)	Size (%)	Origin	Size (%) ^a^
1	1618	10.8	β-sheets (strongly bonded)	37.7
2	1625	33.3	β-sheets (strongly bonded)	4.5
3	1634	3.4	β-sheets (weakly bonded)	7.9
4	1644	19.0	unordered	8.9
5	1651	6.0	α-helices and random coils	8.3
6	1658	7.7	α-helices	3.9
7	1667	14.8	β-turns	22.3
8	1680	4.0	β-sheets (weakly bonded)	4.2
9	1691	0.9	β-turns	2.3

^a^ glycerol-plasticized wheat gluten [39].

**Table 4 polymers-11-00287-t004:** Guggenheim, Anderson and de Boer (GAB) equation parameters.

Peak	Larva	WG 80 °C ^a^	WG 120 °C ^a^
*M* _m_	0.083	0.095	0.096
*C_g_*	1.47	1.359	1.31
*K*	0.953	0.947	0.948
*MRD* (%)	1.32	2.27	6.56

^a^ wheat gluten/glycerol compression molded at 80 °C and 120 °C [47].

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
