# Peer review of "A Protein-Based Material from a New Approach Using Whole Defatted Larvae, and Its Interaction with Moisture"

_polymers, 2019, doi:10.3390/polym11020287_

Round 1
Reviewer 1 Report
The paper presents a new protein-based material is produced from whole defatted larvae and its interaction with moisture. It is a topic of interest to the researchers in the related area, especially material environmental protection field. It is interesting work to be published after the significant revision. My detailed comments are as follows:
1. Infra-red spectroscopy has been used to detect the secondly structure of the new protein material. It is important. But if such kind structure prefer to be useful material for its use, or also appear in other material proteins? More detail results and discussion are needed for the structural analysis. Such analysis is also important for the further study of mechanic of such new material, as the authors stated in the conclusion themselves.
2. The other important part is the ingestions between proteins and water, and many formulas have been used, however, whether these formulas apply to the system in the text is not stated.
In general, after reading the article carefully, I found that all methods used in this paper are almost universal, so the paper lacks some originality. The authors should try to set the problem discussed in this paper more clear, to define the problem and to explain its importance.
Author Response
Comments to reviewer 1
Reviewer comment:
The paper presents a new proteinbased material is produced from whole
defatted larvae and its interaction with moisture. It is a topic of interest to the
researchers in the related area, especially material environmental protection
field. It is interesting work to be published after the significant revision. My
detailed comments are as follows:
Author comment:
We thank the reviewer for the comments.
Reviewer comment:
1. Infrared spectroscopy has been used to detect the secondly structure
of the new protein material. It is important. But if such kind structure prefer to
be useful material for its use, or also appear in other material proteins? More
detail results and discussion are needed for the structural analysis. Such
analysis is also important for the further study of mechanic of such new
material, as the authors stated in the conclusion themselves.
Author comment:
We have now written more clearly in the protein structure part of the R&D section the importance of revealing the secondary structure of the protein and how the observed aggregation of the protein is expected to effect the material properties.
Reviewer comment:
2. The other important part is the ingestions between proteins and water,
and many formulas have been used, however, whether these formulas apply
to the system in the text is not stated.
Author comment:
We do not understand exactly the point made here. The materials we make here are not intended for eating/ingestion, so this issue with ingestion and water/protein has not been investigated here.
Reviewer comment:
In general, after reading the article carefully, I found that all methods used in
this paper are almost universal, so the paper lacks some originality. The
authors should try to set the problem discussed in this paper more clear, to
define the problem and to explain its importance.
Author comment:
The methods are not new, but the material studied has not been reported before. The type of system and approach is also new, where the full larvae-material is used after removal of the fat. We have now more clearly defined the problem and the motivation for the study in the introduction and explained the importance of water-protein interactions to be revealed as well as the secondary structure.

Reviewer 2 Report
The authors report studies on interaction of moisture with proteins extracted from Black soldier fly larvae. Compression molded samples have been characterized and this study merely falls within the aims and scopes of the journal.
The title suggests that a "new" material is being studied. However, proteins from larvae are not new materials and the title should be changed to reflect that.
The novelty of the current study should be spelt out in the introduction.
The paper discusses the proteins as a material but material characteristics (eg. mechanical properties) are not absent.
The proteins have been compression molded at 100 deg C under pressure. Are the proteins stable under these conditions?
What is the intended application of the molded films?
Why are interactions of moisture with the material important? Does moisture have an impact on the end use performance of the material?
If possible, TGA experiments should be performed and results preformed. DSC results should also be presented to determine the plasticization effect of glycerol.
How are the contributions from glycerol treated when analyzing FTIR spectra?
Author Response
Comments to reviewer 2
Reviewer comment:
The authors report studies on interaction of moisture with proteins extracted
from Black soldier fly larvae. Compression molded samples have been
characterized and this study merely falls within the aims and scopes of the
journal.
Author comment:
Proteins are polymers and we contacted the editor before the submission which agreed on the content on water-protein material interactions.
Reviewer comment:
The title suggests that a "new" material is being studied. However, proteins
from larvae are not new materials and the title should be changed to reflect
that.
Author comment:
We agree, there are protein materials from larva, e.g silk, but that is a material produced by the larva and not a material from the larva biomass itself as reported here. We have changed the title and the first sentence in the abstract to better reflect on this.
Reviewer comment:
The novelty of the current study should be spelt out in the introduction.
Author comment:
We have now stated more clearly in the introduction that: This is the first study on using whole defatted larvae for plastic material production.
Reviewer comment:
The paper discusses the proteins as a material but material characteristics
(eg. mechanical properties) are not absent.
Author comment:
Yes, in this study we focus on the interactions between the material and water. As stated in the conclusions the mechanical features will be investigated in a separate study.
Reviewer comment:
The proteins have been compression molded at 100 deg C under pressure.
Are the proteins stable under these conditions?
Author comment:
Yes, we do not see any degradation features in e.g. in IR data. Our experience on many different proteins; gluten, potato protein, pea protein, rape seed protein, shows that 100 °C is not a harmful temperature. The TGA data also shows that what happens between room temperature and 100 °C is the loss of water. However, proteins always unfolds to certain extents during heat treatment, which is usually followed by aggregation (refer to the IR data here).
Reviewer comment:
What is the intended application of the molded films?
Author comment:
A text is now inserted in the conclusions: Based on the results of this work, it is clear that the type of protein material investigated here can only find use in drier climates / indoor applications, separated from liquid water, or in applications where the material is in contact with fatty substances. Possible future applications could be e.g. as covers of lamps, covers of electrical parts/switches and disposable plastic products.
Reviewer comment:
Why are interactions of moisture with the material important? Does moisture
have an impact on the end use performance of the material?
Author comment:
The biggest hurdle for protein materials to be able to replace oil-based plastics is their moisture-sensitivity. It is thus of importance to establish the interactions between moisture and the material. This is now explained in the introduction. The larva material shows slightly lower water uptake at high water activity compared to the gluten material, which is beneficial when using the material in plastics applications. Also, the lower diffusivity of the larva material is beneficial. Protein materials suffer from changes in properties during variations in the relative humidity. It is therefore beneficial to have a lower water diffusivity since the response to a change in relative humidity will be slower and property changes will smoothen out more over time. The effects of lower water uptake and diffusivity is written in the manuscript now.
Reviewer comment:
If possible, TGA experiments should be performed and results preformed.
DSC results should also be presented to determine the plasticization effect of
glycerol.
Author comment:
As suggested by the reviewer we have now performed TGA experiments, and those are included (new Fig. 4).
DSC data is difficult to evaluate in this case, because of the overlapping effects of water (freezing, melting, evaporation). The films are soft at room temperature, which indicate a Tg below room temperature. On the other hand, we have now also added calorimeter data to the TGA data in the new Fig. 4.
Reviewer comment:
How are the contributions from glycerol treated when analyzing FTIR
spectra?
Author comment:
Glycerol contributes with a small shoulder in the amide I region which will, if at all, contribute to the Amide I IR absorption region at higher wavenumbers, which if removed from the IR spectrum would lead to a curve showing even more protein aggregation then presented here. However, this effect is small and in the comparisons of the secondary structure to the other protein (Table 3), it is with the same glycerol content.
Round 2
Reviewer 1 Report
The current version has included more structural analysis. It can be accepted afther a minor revision. Can authors explain physical or biological meaning of each mathematical formula? I think that it would make the readers of polymers understand easier.
Author Response
It is difficult to explain some of the equations more than is already there. However, we have explained equations further where we felt it was possible (in red).
Reviewer 2 Report
The authors have attended to comments and clarifications raised by the reviewer in the first review. The paper has thus been significantly improved and may now be accepted for publication.
Author Response
Dear reviewer, many thanks.